# Photoinduced Antibacterial Activity and Cytotoxicity of CdS Stabilized on Mesoporous Aluminosilicates and Silicates

**DOI:** 10.3390/pharmaceutics14071309

**Published:** 2022-06-21

**Authors:** Anna Stavitskaya, Eliza Sitmukhanova, Adeliya Sayfutdinova, Elnara Khusnetdenova, Kristina Mazurova, Kirill Cherednichenko, Ekaterina Naumenko, Rawil Fakhrullin

**Affiliations:** 1Functional Aluminosilicate Nanomaterials Lab, Gubkin University, 119991 Moscow, Russia; eliza.sit@mail.ru (E.S.); sayfutdinova.a@gubkin.ru (A.S.); husnetdenova.1998@mail.ru (E.K.); mazurovachris55@mail.ru (K.M.); cherednichenko.k@gubkin.ru (K.C.); kazanbio@gmail.com (R.F.); 2Bionanotechnology Lab, Institute of Fundamental Medicine and Biology, Kazan Federal University, Kreml uramı 18, 420008 Kazan, Republic of Tatarstan, Russia; ekaterina.naumenko@gmail.com

**Keywords:** halloysite, MCM-41, SBA-15, *S. aureus*, visible light, quantum dots, photocatalysis

## Abstract

Inactivation of bacteria under the influence of visible light in presence of nanostructured materials is an alternative approach to overcome the serious problem of the growing resistance of pathogenic bacteria to antibiotics. Cadmium sulfide quantum dots are superefficient photocatalytic material suitable for visible light transformation. In this work, CdS nanoparticles with size of less than 10 nm (QDs) were synthesized on the surface of natural and synthetic mesoporous aluminosilicates and silicates (halloysite nanotubes, MCM-41, MCM-41/Halloysite, SBA-15). Materials containing 5–7 wt.% of CdS were characterized and tested as agents for photocatalytic bacteria degradation of Gram-positive *S. aureus* and Gram-negative *E. coli* with multiple antibiotic resistance. Eukaryotic cell viability tests were also conducted on the model cancer cells A 459. We found that the carrier affects prokaryotic and eukaryotic toxicity of CdS quantum dots. CdS/MCM-41/HNTs were assumed to be less toxic to eukaryotic cells and possess the most prominent photocatalytic antibacterial efficiency. Under visible light irradiation, it induced 100% bacterial growth inhibition at the concentration of 125 μg/mL and the bacteriostatic effect at the concentration of 63 μg/mL. CdS/MCM-41/HNTs showed 100% *E. coli* growth inhibition in the concentration of 1000 μg/mL under visible light irradiation.

## 1. Introduction

Due to the growing resistance of microorganisms to antibiotics, there is a need for new approaches to solve the problem. Exploring alternative methods of bacterial cell degradation is one of the most important tasks to ensure the sanitary and hygienic safety of society [1]. Light stimulated destruction of prokaryotic cells in presence of photoactive nanomaterials is a promising direction with wide application possibilities [2]. Several types of nanomaterials are in a focus of attention such as inorganic and organic nanoparticles, layered structures, photosensitive metal complexes [3,4,5,6]. Depending on the composition and structure, nanomaterials may inactivate bacteria under the influence of ultraviolet, visible or infrared irradiation.

Recent studies showed that transition metal sulfides with adjustable spectral characteristics and controlled morphology are very promising bactericidal agents capable of causing severe oxidative stress in bacteria under radiation [7,8,9]. At the same time, the development of methods for stabilizing nanoparticles is still relevant. Reducing possible toxic effects on eukaryotic cells and organisms in general is a priority area of research, as well as a limiting factor in the implementation of many technologies.

CdS is an efficient photocatalyst as well as potential antibacterial agents [10,11,12]. Photocatalytic activity of cadmium sulfide nanoparticles is strongly dependent on the particle size, shape and composition [13]. With the growth in particle size, the efficiency of the reactions may be decreased due to recombination of the photogenerated electron-hole pair in the bulk of material. That is why quantum dots are much more efficient that other nanostructures as they supply the highest surface area and the lowest possible volume of the particle. The antibacterial activity of semiconductors is associated with the production of reactive oxygen species (ROS) that are sufficiently activated under irradiation [14]. The ability to produce ROS has a dominant effect on antibacterial efficacy of CdS-based compounds. Antibacterial activity can rather be tuned by doping with metals or shape and size modifications [9,15]. 

Recent studies demonstrate the potential of abundantly available halloysite clay nanotubes for life science applications, from drug delivery via oral or topical administration, to tissue scaffolds and regenerative medicine, while assessing their cellular internalization, stability, biosafety and biocompatibility are featured [16,17]. The production of hierarchical carriers using halloysite as a template is a promising tool to vary its physical and catalytic properties [18]. Here, we aim to learn how the carrier and its modification may influence the ability of mesoporous silicates and aluminosilicates with CdS quantum dots to degrade antibiotic resistant bacteria. This is important for further investigation of inorganic mesoporous materials for biomedical applications such as synthesis of multifunctional materials and drug delivery systems.

In this work, we studied for the first time a photocatalytic degradation of Staphylococcus aureus (*S. aureus*) and Escherichia coli (*E. coli*) using CdS NPs stabilized on different mesoporous silicates (MCM-41, SBA-15) and aluminosilicates. Aluminosilicates presented in the study are natural aluminosilicate nanotubes (HNTs) and recently developed hierarchic mesoporous carrier MCM-41/HNTs. MCM-41/HNTs has never been studied for biomedical applications before. Toxicity of CdS stabilized on mesoporous inorganic carriers towards A549 cancer cells was investigated for the first time. It was assumed that CdS/MCM-41/HNTs was the most active against both Gram-positive and negative bacteria under visible light as well as less toxic to A 549 cancer cells.

## 2. Materials and Methods

### 2.1. Materials

Halloysite (Al_2_Si_2_O_5_(OH)_4_·2H_2_O) (HNTs) cadmium nitrate tetrahydrate (Cd(NO_3_)_2_·4H_2_O) (Sigma-Aldrich, St. Louis, MO, USA), thioacetamide (C_2_H_5_NS), ammonium hydroxide (NH_4_OH), cetyltrimethylammonium bromide (CTAB) (C_19_H_42_BrN), Pluronic (P123), (3-mercaptopropyl)-methyl-dimethoxysilane (RSH), (3-aminopropyl)triethoxysilane (APTES) were purchased from Sigma-Aldrich (St. Louis, MO, USA), tetraethyl orthosilicate (C_2_H_5_O)_4_Si, (TEOS, 98%), ethanol (C_2_H_5_OH, 96%) were purchased from EKOS-1 (Staraya Kupavna, Russia), hydrochloric acid (HCl, 37%) was purchased from Sigma Tech (Moscow, Russia).

### 2.2. MCM-41 and MCM-41/HNTs Synthesis

A total of 20 mL of aqueous CTAB (1.312 g) solution was added to 123 mL of NaOH (0.143 g) aqueous solution. 8.62 g of tetraethoxysilicate (TEOS) was added dropwise to the above solution (2 drops per minute) and pH was adjusted to 10–11. The mixture was stirred at room temperature for 24 h. The mixture was then heated to 90 °C and kept at this temperature overnight without stirring, filtered, washed with distilled water and dried at room temperature for 24 h. Then, put into a muffle furnace at 100 °C for 24 h and calcined in air stream at 550 °C for 4 h.

During the synthesis of MCM-41/HNTs the calculated amount of halloysite nanotubes were added to CTAB solution in order to obtain MCM-41:HNTs = 1. The further synthesis steps were the same.

### 2.3. SBA-15 Synthesis

The SBA-15 carrier was synthesized using Pluronic P123 block copolymer, which was dissolved in 143 mL of distilled water and 19.5 mL of 37% HCl solution with constant stirring at 40 °C for 5 h. Then, 8.62 g of TEOS was added to the resulting solution and stirred for 20 h. The mixture was then heated to 90 °C and kept at this temperature overnight without stirring. After filtration and washing with distilled water, the white precipitate was dried at 70 °C, and then calcined in an air stream at 550 °C for 5 h.

### 2.4. Modification of HNTs and SBA-15

To obtain SBA-15-SH, the resulting SBA-15 carrier was dispersed in ethanol in an ultrasonic bath for 30 min. (3-mercaptopropyl)-methyl-dimethoxysilane was added dropwise with constant stirring to an alcohol solution of SBA-15 (in a ratio of 1 g per 0.2 mL of silane). Then, the temperature was raised to 60 °C and stirred during the day. Then, the mixture was centrifuged, washed with ethanol three times and dried at 60 °C.

To modify HNTs, a similar procedure was applied using (3-aminopropyl)triethoxysilane.

### 2.5. Synthesis of CdS

CdS nanoparticles were synthesized on the surface of mesoporous carriers as follows: cadmium nitrate ethanol solution was added to dispersion of carrier in ethanol and sonicated for 30 min. Cd(NO_3_)_2_·4H_2_O concertation was calculated to obtain Cd of 5 wt.%. Afterwards, a solution of thioacetamide (S/Cd = 1) was added dropwise to a dispersion and stirred for 10 min followed by NH_3_·H_2_O (1 mL) adjusting. As a result, yellow precipitate was formed, separated by centrifugation, washed several times with ethanol and dried at 65 °C.

### 2.6. Samples Characterization

The structure and morphology of the prepared samples were studied using a transmission electron microscope (TEM) JEM-2100 UHR (JEOL, Tokyo, Japan) with a magnification factor of 50–1,500,000 times and an image resolution of 0.19 nm at 200 kV. 

Scanning transmission electron microscopy (STEM) in combination with energy dispersive X-ray spectroscopy (EDX) was employed to map the distribution of elements in samples using JEM-2100 UHR instrument (JEOL, Tokyo, Japan). 

The elemental composition was determined using inductively coupled plasma emission spectroscopy ICPE-9000. 

X-ray diffraction (XRD) was carried out using a SmartLab instrument (Rigaku, Tokyo, Japan) with monochromatic Cu-Kα-radiation (λ = 1.5418 Å) in 0–80° 2θ angle range at 5°/min rate.

The zeta potential of the samples was measured in the pH = 7.0 using the SZ-100 system (Horiba, Tokyo, Japan) at the electric field strength of 39 V × m^−1^.

The UV-Vis diffuse reflectance spectra were recorded in the range of 400–800 nm with 1 nm resolution using UV-Vis spectrophotometer (Scimadzu, Tokyo, Japan) equipped with a diffuse reflectance accessory. The UV-Vis spectra were recalculated in the Kubelka-Munk coordinates as follows:F(R) = (1 − R/100)^2^/(2R/100)(1)
where R is the reflectance (%). The optical band gap (Eg) for the synthesized photocatalysts was estimated using the Tauc model for direct allowed transitions by plotting (F(R) × hν)^2^ versus hν followed by a linear extrapolation to intercept the energy axis.

### 2.7. Antibacterial Assay

The clinical isolate of strains 119 (*S. aureus* MSSA) and 3 (*E. coli*) was used for antimicrobial assay. A suspension of *S. aureus* or *E. coli* was prepared by the colony suspension method: three to five colonies from a nonselective nutrient agar medium incubated at 37 °C for 18 h were taken by a loop and moved to a sterile 0.9% sodium chloride solution. The suspension was adjusted to produce turbidity equivalent to 0.5 McFarland standard, corresponding to approximately 1.5 × 10^8^ CFU × mL^–1^ of bacteria. Then, the obtained suspension was diluted to produce bacterial density equal to 1.5 × 10^6^ CFU × mL^–1^. The tested sample in calculated concentration was suspended in 1.0 mL of Mueller–Hinton Broth (Becton Dickinson, Franklin Lakes, NJ, USA). The resulting concentration of the sample in the wells varied from 63 to 1000 μg/mL. After 100 μL of the suspension was added to the microplate wells and then the wells were inoculated with 100 μL of the bacterial suspension. Microplates were incubated at 37 °C for 18 h. After incubation, the material from each well was seeded on Muller Hinton Agar (Becton Dickinson, Franklin Lakes, NJ, USA) for the quantitative counting of surviving cells using 10-fold dilution series. Petri dishes were incubated at 37 °C for 18 h After incubation, the grown colonies were counted, and the number of CFUs was calculated.

### 2.8. Cell Viability Tests

The human adenocarcinoma cells (A549) from American Type Culture Collection (ATCC, Manassas, VA, USA) were routinely maintained in Dulbecco’s Modified Eagle Medium (DMEM) (Sigma) culture medium supplemented with 10% (*v*/*v*) fetal bovine serum (PAA) and 100 U/mL penicillin and 100 μg/mL streptomycin at 37 °C in a humidified atmosphere with 5% CO_2_. After 24 h of cultivation, cells were treated with samples in studied concentrations (100, 200, or 300 μg/mL).

To determine the viability of cells MTT-test based on metabolic activity of cells was used. 3-(4,5-dimethylthiazol-2-cyl)-2,5-di-phenyltetrazolium bromide (MTT) (Sigma-Aldrich, St. Louis, MO, USA) was used at a concentration of 5 mg/mL in phosphate buffer. Working solution of MTT was added to the culture medium in a ratio of 1:10 after 24 h incubation with samples. 

Cell cultures were incubated with MTT for 4–5 h until the formazan crystals formed inside cytoplasm under influence of intracellular enzymes. Formazan was extracted using DMSO (Sigma-Aldrich, St. Louis, MO, USA). The experiment was carried out in six replications. The optical density of formazan was recorded at 544 nm using a Multiscan microplate photometer (Multiskan FC, Thermo Fisher Scientific, Waltham, MA, USA).

Cell viability based on membrane-integrity of A549 cells after incubation with samples was determined using a kit for differential staining of living and dead cells’ nuclei (blue live/green dead) (Life Technologies, Thermo Fisher Scientific, Waltham, MA, USA) according to the protocol provided by the manufacturer. Fluorescent images were obtained using Carl Zeiss Axio Imager Z1 microscopy (Jena, Germany). The percentage of dead or membrane-compromised cells in each sample was calculated by analyzing at least 200 cells in different fields of view.

## 3. Results

### 3.1. Structure and Morphology

The structural investigations were carried out with help of XRD analysis (Figure 1). Crystalline structure of halloysite nanotubes in the sample CdS/HNTs was revealed. The characteristic signals on low angles proved the preservation of MCM-41 and SBA-15 pore structure after CdS synthesis in CdS/MCM-41, CdS/MCM-41-HNTs, CdS/SBA-15, CdS/SBA-15-SH (Figure 1). Signals correspondent to CdS at 26.7°, 44.1°, 52.2° were detected on refractograms of all samples except for CdS/SBA-15-SH. In the case of CdS/HNTs, a reliable signal at 44.1° was observed [19].

Crystalline CdS was also detected in all samples using high resolution TEM (Figure 2). According to TEM an average size of CdS nanoparticles grafted to HNTs modified with aminosilane was 6 ± 1 nm. CdS quantum dots were also formed on mesoporous MCM-41 and hierarchical material MCM-41/HNTs. Particles were evenly distributed over the surface of the carriers. For the CdS/SBA-15 sample, the predominant formation of nanoparticles on the surface of the carrier was observed. In some places, CdS nanoparticles with size of more than 10 nm with a flowerlike shape were observed. Silanization of mesoporous silicon oxide SBA-15 (CdS/SBA-15-SH) stabilized nanoparticles and their average diameter decreased to 4 ± 1 nm. The formation of rod shape particles was also revealed (Figure 2). 

According to the results of electron microscopy investigations (Figure 2 and Figure 3A) all samples have different shape and size of the carriers. The HNTs (in CdS/HNTs) length varies from 200 to 700 nm and diameter is from 7 to 15 nm. CdS/MCM-41 was characterized by MCM-41 microspheres and amorphous MCM-41 with particle size from 600 nm to 1.2 μm. In CdS/MCM-41/HNTs the morphology of carrier represented a composite where MCM-41 was formed mostly on the surface of clay nanotubes and the size of such particles was 200–700 nm in length and 40–200 nm in width. SBA-15 and SBA-15-SH were characterized by mostly cylindric particles with length of up to 1.2 μm (Figure 2a,e). Elemental mapping confirms the presence of such elements as: Si (red), Al (orange) and Cd (green) in all samples (Figure 2b). Most uniformed distribution of CdS was observed in the samples CdS/MCM-41 and CdS/SBA-15.

Cd content based on ICP-EOS data as well as zeta-potential values of CdS/HNTs, CdS/MCM-41, CdS/MCM-41-HNTs, CdS/SBA-15, CdS/SBA-15-SH are in Table 1. Surface area of CdS/HNTs, CdS/MCM-41-HNTs and CdS/SBA-15-SH revealed by BET is in Table 1.

### 3.2. Spectral Characteristics

IR-spectra of all samples were analyzed (Figure 4). The bands at 960 and 1090 cm^−1^ corresponding to Si–OH vibration and Si–O–Si asymmetric stretching vibration of SiO_2_ were observed in all silicate-based samples as well as CdS/MCM-41 [20]. IR analysis of CdS/HNTs revealed the common IR-spectrum for halloysite nanotubes with 1107 cm^−1^ assigned to apical Si–O as well as at 1030 and 691 cm^−1^ assigned to vibrations of Si–O–Si. CdS presence as well as surface modification of HNTs or SBA-15, however, could not be reliably established from FT-IR spectroscopy data.

Halloysite as well as SiO_2_ (band gap of 8–11 eV) shows almost 100% reflectance in the region from 250 nm to 800 nm. The adsorption band edge at 520–570 nm for CdS containing samples suggests the presence of CdS particles of different size. All UV-Vis diffuse reflectance spectra had broad absorption region below 550 nm that is attributed to the charge transfer from the valence to the conduction band of CdS (Figure 5a). CdS band gap was determined by standard method by plotting (F(R) × hν)^2^ for direct semiconductors versus hν, followed by linear extrapolation to intercept the energy axis [21]. Bang gap energy calculated using Tauc method equal to 2.36 eV for CdS/MCM-41 which is close to bulk CdS (2.4 eV), 2.45 eV for CdS/SBA-15, 2.47 eV for CdS/SBA-15-SH, 2.54 eV for CdS/MCM-41-HNTs and 2.58 eV for CdS/HNTs (Figure 5b).

### 3.3. Antibacterial Activity Againts S. aureus

The antibacterial activity of CdS nanoparticles stabilized on different mesoporous carriers was studied using Gram-positive *S. aureus* and Gram-negative *E. coli.* The antibacterial tests were conducted using different composites concentrations with and without visible light illumination (Figure 6). 

Antibacterial activity in the dark was not pronounced. CdS/SBA-15-SH was the most toxic to *S. aureus* in the concentration of 1000 μg/mL (Figure 6a).

*S. aureus* degradation was much more efficient under visible light irradiation (Figure 6b). CdS/HNTs showed 100% bacterial growth inhibition at the concentration of 1000 μg/mL. For all other samples 100% bacterial growth inhibition was observed for the concentrations in 125–1000 μg/mL range. In some cases, even a small concentration of composite demonstrated strong bactericidal effect. Thus, for instance, 63 μg/mL CdS/MCM-41/HNTs and CdS/SBA-15 (ca. 3.8 μg/mL of CdS) showed 50% *S. aureus* growth inhibition efficiency under visible light.

Antibacterial test on *E. coli* showed that only two samples inhibited the growth of *E. coli* under irradiation in the concentration of 1000 μg/mL: CdS/MCM-41/HNTs and CdS/SBA-15-SH. *E. coli* growth inhibition reached 100% for both samples.

### 3.4. Cell Viability

To access possible toxicity effects on mammalian cells, cytotoxicity tests were conducted on the A549 cells. Human lung carcinoma cell culture A549 is widely used as a model for studying the toxicity of various classes of compounds, including nanomaterials. In addition, this cell type has a number of metabolic pathways that ensure the insensitivity of these cells to toxicants, if materials exhibit a toxic effect on this model, then this effect will be even more pronounced on normal non-malignant cells [22,23].

The enzymatic activity of cells was assessed after incubation with nanocomposites by MTT-test (Figure 7). This test is based on the ability on the succinate dehydrogenase to reduce MTT to the corresponding formazan [24]. The amount of formazan formed in the cells is proportional to the number of metabolically active cells and can be easily analyzed spectrophotometrically.

Enzymatic activity of A549 cells decreased in dose-dependent manner with an increase in the composite’s concentration. Most pronounced toxic effect was showed for nanoparticles deposited on halloysite modified with aminosilane (CdS/HNTs) and SBA-15 modified with mercaptosilane (CdS/SBA-15-SH) (Figure 7). These results were correlated with Live/dead staining for CdS/SBA-15-SH where 100% cells’ nuclei were stained with green component of kit (Figure 8) and cells were determined as dead or membrane-compromised. Only 63% of cells treated with sample CdS/HNTs were as dead/membrane compromised [25]. Incubation of cells with the samples CdS/SBA-15 and CdS/MCM-41/HNTs at the concentration of 100 μg/mL did not lead to the decrease in the enzymatic activity of cells. These parameters were at the control level. However, the membranes were damaged as demonstrated in the Live/dead staining (Figure 8). For the sample CdS/MCM-41 the correlation between the membrane integrity losing and decreasing of the enzymatic activity was observed.

## 4. Discussion

Successful synthesis of CdS in all studied samples was proven by XRD and TEM analysis. Absence of CdS Bragg’s diffraction peaks in CdS/SBA-15-SH (Figure 1) could be explained by very small particles size of CdS [19]. All of the sample were characterized by close (4.1–4.7 wt.%) Cd content, which is close to calculated value (5 wt.%) and correspondent to CdS concentration of 5–7 wt.%. This is quite low compared with other studies [2,6,8]. The lowest Cd concentration (4.1 wt.%) was found in CdS/HNTs and the highest Cd concentration of 4.7 wt.% is attributed to CdS/SBA-15-SH. Difference in Cd concentration was most likely due to different adsorption capacities of samples. FT-IR spectroscopy analysis provided the information on the formation of silicates and preservation of halloysite during the synthesis.

Calculation of band gap energy using Tauc method is a common way to identify the formation and nature of semiconductor. Decrease in a bang gap energy of CdS deposited on mesoporous materials compared to bulk CdS parameters showed the formation of nanoparticles with size of less than 10 nm [26]. This feature may be ascribed to the well-known quantum-confinement effect [27,28]. A band gap value of 2.36 eV for CdS/MCM-41, close to bulk CdS (2.4 eV), may be evidence of close package of nanoparticles in the sample. TEM of this sample (Figure 2a,b) showed that CdS nanocrystals were located close to each other, forming a kind of CdS film. It should be mentioned that the carrier modification also played significant role in the particles size and as a result physical properties of nanomaterials. Band gap values of CdS/SBA-15 (2.45 eV) and CdS/SBA-15-SH (2.47 eV) is another evidence of decrease in CdS particles size after SBA-15 modification due formation of additional adsorption sited on the carrier. The band gaps of 2.54 eV in case of CdS/MCM-41-HNTs and 2.58 eV in case of CdS/HNTs indicated partial oxidation of CdS on the surface of nanocrystals to CdO (band gap of 2.52 eV).

Antibacterial activity of nanomaterial in the dark and toxicity to A549 is mostly associated with the release of Cd ions [29]. The rate of Cd release into the media should grow with decrease in CdS particles size. The data analysis did not reveal liner correlation between Cd concentration and particles size of CdS and toxicity of nanomaterials. It seems that carrier morphology and surface properties of samples play more significant role.

Photocatalytic bacterial degradation under visible light was much more pronounced for both Gram-positive and negative model organisms. Activity of samples against *S. aureus* could be place in the row: CdS/MCM-41/HNTs, CdS/SBA-15 > CdS/MCM-41, CdS/SBA-15-SH > CdS/HNTs. *E. coli* degradation under visible light was not as efficient due to its membrane structure. CdS/MCM-41/HNTs and CdS/SBA-15-SH were more efficient than the rest of the materials.

Based on electron microscopy studies of CdS/MCM-41 as well as its spectral characteristics one might expect high possibility of electron-holes recombination due to close position of CdS nanoparticles and as a result decrease in photocatalytic activity. Previously it was shown that CdS/MCM-41 was less active in photocatalytic hydrogen generation under visible light compared to CdS/MCM-41/HNTs [21]. Here, this correlation was also observed. The presence of additional Al atoms in CdS/MCM-41/HNTs made charge separation more efficient which increases photocatalytic activity and as a result higher concentration of ROS may be achieved. CdS/SBA-15 was also very efficient against *S. aureus* due to high porosity and large pores size of the carrier (Table 1). Optimal size of CdS and its homogeneous distribution on the surface of the silicate promoted photocatalytic activity. 

Higher resistance of *E. coli* to membrane damage by ROS need a development of more efficient and stable photocatalysts. In contrast to other samples, CdS/MCM-41/HNTs was efficient in both killing Gram-positive and negative bacteria. Photocatalytic inhibition of *E. coli* reached 100% in a concentration of 1 mg/mL; bacteriostatic effect was observed in the concertation of 63 µg/mL on *S. aureus*. All this shows that CdS/MCM-41/HNTs was the most efficient material in our study.

Samples toxicity to A 549 cells was mainly due to the fact that Cd^2+^ ions could release in media and affect cells. Leakage of cadmium can be associated with the penetration of oxygen and protons to QDs [30]. Released Cd^2+^ ions can bind to sulfhydryl groups of mitochondrial proteins, which leads to cell toxicity [31]. This can manifest itself both in apoptotic changes in the cell and in violation of the enzymatic activity of mitochondria. Lopez et al. reported that in the presence of serum, concentrations of Cd^2+^ lower than 10 nmol/mL did not induce necrotic cell death but apoptotic cell death in cortical neurons [32].

Differences in the results of MTT and live/dead staining in our experiments can be explained by the fact that cells remain metabolically active despite membrane damage. Therefore, it is important to apply an integrated approach to characterize the toxic effect and study both the enzymatic activity of cells and their permeability.

## 5. Conclusions

CdS quantum dots were synthesized on the surface of natural and synthetic silicates and aluminosilicates. MCM-41, SBA-15, halloysite and MCM-41/HNTs were chosen as carriers to study the effect of carrier on photocatalytic degradation of *S. aureus* and *E. coli* under the action of visible light. Concentration of CdS in the samples was 5–7 wt.%. No pronounced activity of highly concentrated composite (1000 μg/mL) was detected in the dark. Meanwhile under visible light irradiation even a comparatively small concentrations of all samples (e.g., 125 μg/mL) except for halloysite-based material showed 100% bacterial growth inhibition. Two out of five studied composites (CdS/MCM-41/HNTs and CdS/SBA-15) showed bacteriostatic effect at the concentrations of 63 μg/mL. Two out of five samples CdS/MCM-41/HNTs and CdS/SBA-15-SH showed 100% growth inhibition of *E. coli* in concentration of 1 mg/mL under action of visible light.

We assume that among studied antibacterial agents less toxic to A549 and at the same time the most efficient as photocatalytic antibacterial agent was CdS/MCM-41/HNTs. SBA-15 is also perspective carrier for antibacterial nanoparticles stabilization. However, due to expensive chemicals used for SBA-15 production MCM-41/HNTs is more preferable.

## Figures and Tables

**Figure 1 pharmaceutics-14-01309-f001:**
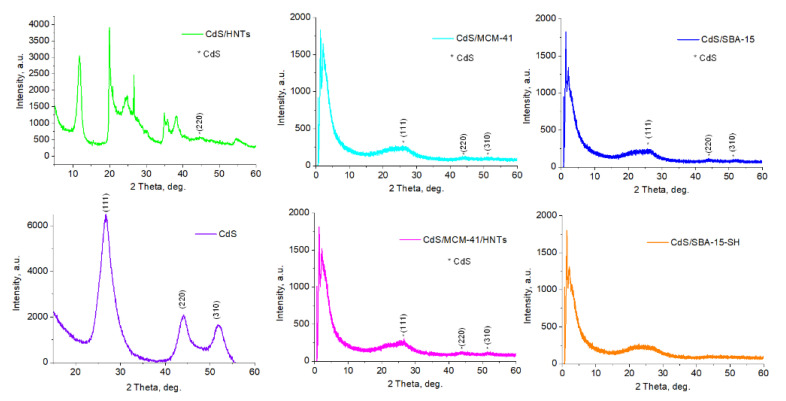
XRD of CdS, CdS/HNTs, CdS/MCM-41, CdS/MCM-41-HNTs, CdS/SBA-15 and CdS/SBA-15-SH.

**Figure 2 pharmaceutics-14-01309-f002:**
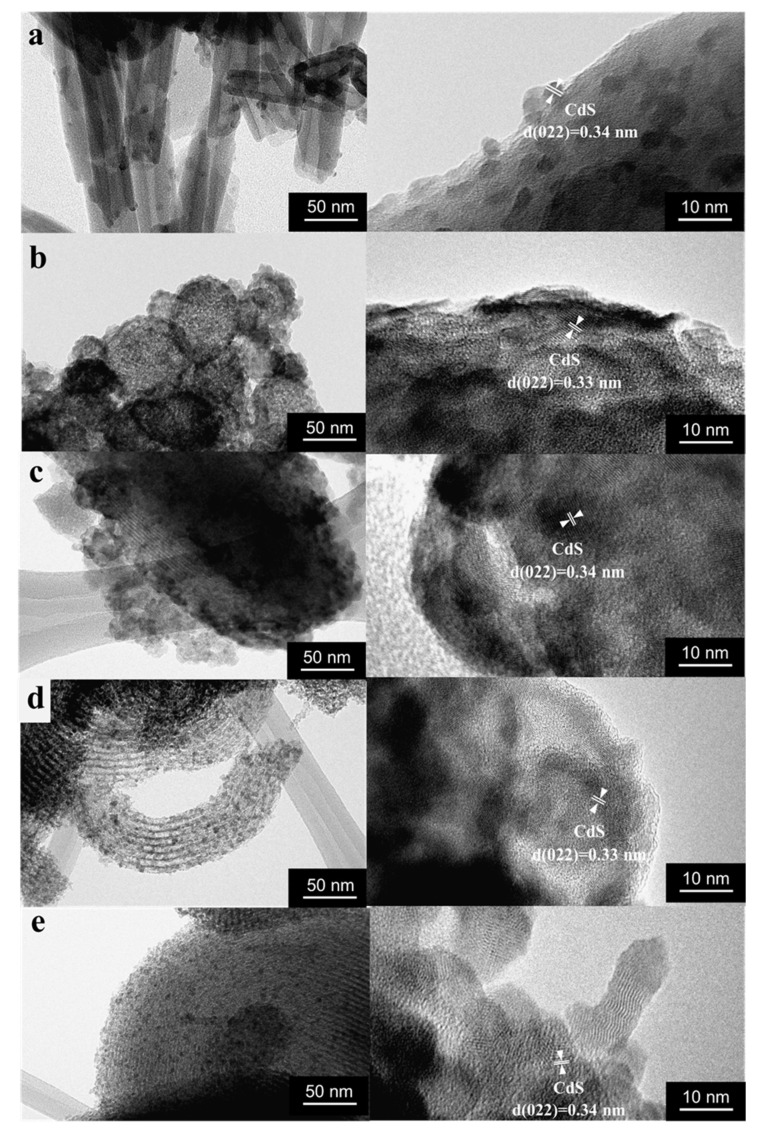
TEM under different magnifications of CdS/HNTs (**a**), CdS/MCM-41 (**b**), CdS/MCM-41-HNTs (**c**), CdS/SBA-15 (**d**), CdS/SBA-15-SH (**e**).

**Figure 3 pharmaceutics-14-01309-f003:**
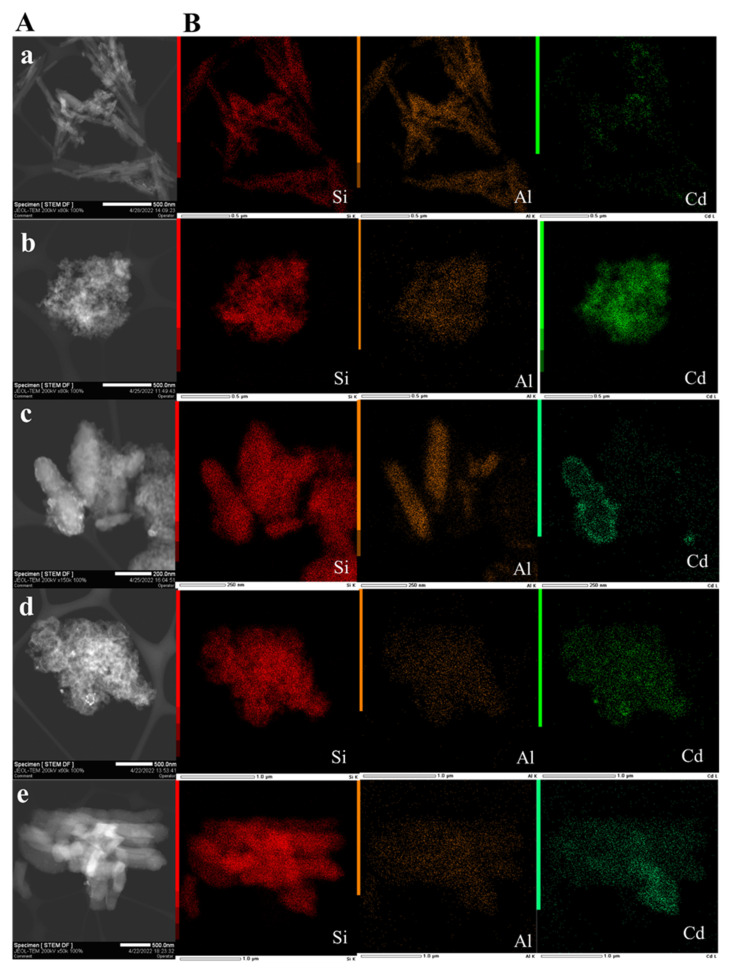
STEM (**A**) and elemental mapping (**B**) of CdS/HNTs (**a**), CdS/MCM-41 (**b**), CdS/MCM-41-HNTs (**c**), CdS/SBA-15 (**d**), CdS/SBA-15-SH (**e**).

**Figure 4 pharmaceutics-14-01309-f004:**
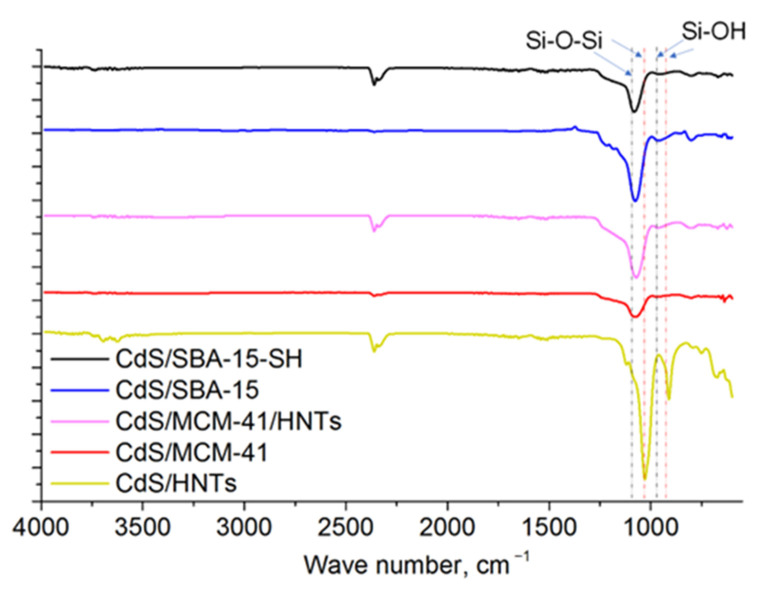
FT-IR spectra of CdS-based mesoporous materials.

**Figure 5 pharmaceutics-14-01309-f005:**
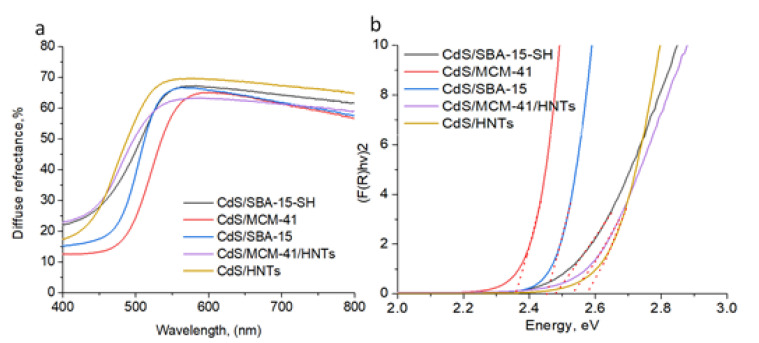
Diffuse reflectance spectra (**a**) and correspondent Tauc plot (**b**) of CdS-based mesoporous materials.

**Figure 6 pharmaceutics-14-01309-f006:**
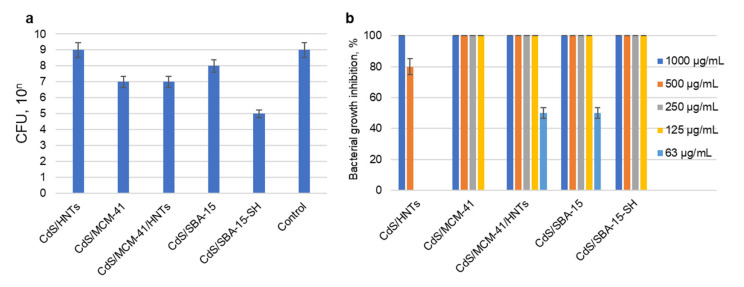
Antibacterial activity of CdS nanocomposites against *S. aureus*: in the dark (concentration of composite 1000 μg/mL) (**a**); under visible light irradiation depending on the concentration of composite (63–1000 μg/mL) (**b**).

**Figure 7 pharmaceutics-14-01309-f007:**
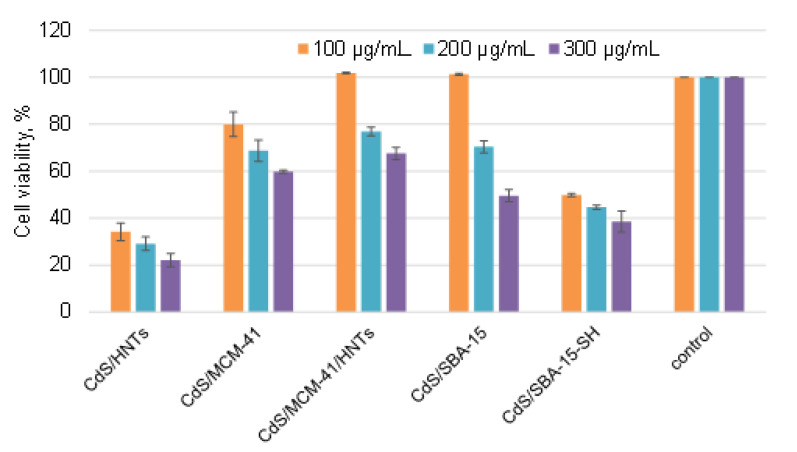
A549 cells metabolic activity analyzed using MTT assay after treatment with increasing concentrations of composites Live/dead staining of A549 cells treated with samples.

**Figure 8 pharmaceutics-14-01309-f008:**
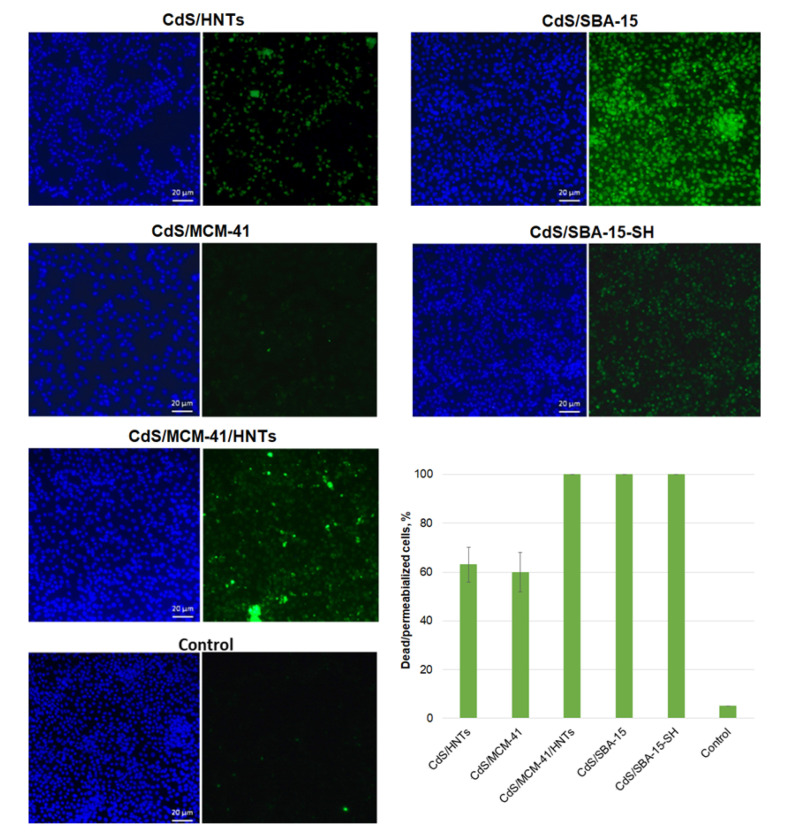
Dead or membrane-compromised cells’ nuclei stained as green, total cells’ nuclei counterstained with DAPI.; Concentration of nanocomposites 100 μg/mL; scale bar—20 μm.

**Table 1 pharmaceutics-14-01309-t001:** Physico-chemical characteristics of CdS stabilized on mesoporous aluminosilicates and silicates.

Sample	Cd Content, wt.% (ICP-EOS)	Zeta–Potential, mV	S_BET_, m^2^/g
CdS/HNTs	4.1	−20	67
CdS/MCM-41	4.6	−34	-
CdS/MCM-41-HNTs	4.3	−50	746
CdS/SBA-15	4.5	−45	845
CdS/SBA-15-SH	4.7	−48	-

## Data Availability

Not applicable.

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
