# Peer review of "Photoinduced Antibacterial Activity and Cytotoxicity of CdS Stabilized on Mesoporous Aluminosilicates and Silicates"

_pharmaceutics, 2022, doi:10.3390/pharmaceutics14071309_

Round 1
Reviewer 1 Report
The authors synthesized CdS nanoparticles on the surface of natural and synthetic mesoporous aluminosilicates and silicates (halloysite nanotubes, MCM-41, MCM-41/Halloysite, SBA-15). The author characterized CdS as agents for photocatalytic bacteria degradation and compatibility with cell. It is found that the carrier for quantum dot synthesis matters. CdS/MCM-41/HNTs were assumed to be less toxic to eukaryotic cells and possess the most prominent photo-catalytic antibacterial efficiency.
The authors addresses the scientific problem nicely and provide sufficient data to support the observation. I recommend to publish.
Author Response
Thank you
Reviewer 2 Report
In this manuscript, authors synthesized CdS nanoparticles with size of less than 10 nm (QDs) on the surface of natural and synthetic mesoporous aluminosilicates and silicates for photocatalytic bacteria degradation of Gram-positive S. aureus. It is not recommended for publication in pharmaceutics. Please find detailed comments as below:
- The research is not innovative and the experiment is not sufficient and complete.
- Please provide more discussion about all experimental result as it is authors explain too little to make it comprehendible.
- The format of the references is not uniform. In some reference titles, only the first letter of the first word is capitalized, and in some references, the first letter of all words is capitalized. Even some references do not have a year. So you need to carefully check the format of the revised reference.
Author Response
- The research is not innovative and the experiment is not sufficient and complete.
Investigation of antibacterial efficiency and toxicity to A 459 of CdS stabilized on mesoporous silicates (MCM-41, SBA-15) and aluminosilicates including novel MCM-41/ HNTs and halloysite nanotubes was conducted for the first time. Here we show efficiency of photocatalytic bacterial degradation using CdS QDs stabilized on carriers. We completed the experimental with XRD, FT-IR, surface area analysis and showed results of antibacterial tests both for S. aureus and E. coli. For the first time we present an efficiency of MCM-41/HNTs as a carrier for antibacterial nanoparticles.
- Please provide more discussion about all experimental result as it is authors explain too little to make it comprehendible.
We modified discussion section (page 11-12).
- The format of the references is not uniform. In some reference titles, only the first letter of the first word is capitalized, and in some references, the first letter of all words is capitalized. Even some references do not have a year. So you need to carefully check the format of the revised reference
We corrected references section.
Reviewer 3 Report
It was a manuscript about the fabrication and evaluation of the antibacterial and cytotoxicity features of CdS stabilized on mesoporous aluminosilicates and silicates. Here are some comments on this study that should be considered before publication:
- Please check the whole text, there are some grammatical mistakes in the text that should be corrected.
- Please rewrite material section.
- Please explain about the results of figure 1B, STEM results.
- “Cd content revealed by ICP-EOS was 4.6 wt% for CdS/MCM-41, 4.5 wt% for CdS/SBA-15, 4.7 wt% for CdS/SBA-15-SH, 4.3 wt % for CdS/MCM-41-HNTs, 4.1 wt% for CdS/HNTs. The Zeta-potential values of CdS/HNTs, CdS/MCM-41, CdS/MCM-41-HNTs, CdS/SBA-15, CdS/SBA-15-SH are -20, -34, -50, -45, -48 mV respectively” please explain more about these results. Moreover, please write the results of Zeta potential tests in a table.
- “. Bang gap energy calculated using Tauc method equal to 2.36 eV for CdS/MCM-41 which is close to bulk CdS (м.б. написать сколько это?), 2.45 eV for CdS/SBA-15, 2.47 eV for CdS/SBA-15-SH, 2.54 eV for CdS/MCM-41-HNTs and 2.58 eV for CdS/HNTs.” please explain about these results more. What did they show?
- Authors didn’t mention the results of XRD tests.
- “For all other samples 100% bacterial growth inhibition was observed for the concentrations in 125-1000 µg/mL range. In some cases, even a small concentration of composite demonstrated strong bactericidal effect. Thus, for instance, 63 µg/mL CdS/MCM-41/HNTs and CdS/SBA-15 (ca. 3.8 µg/mL of CdS) showed 50% S. aureus growth inhibition efficiency under visible light.” why it is like that? What does this show?
- Please redraw the curve of MTT assay based on the cell viability percentage and explain the results of that.
- “Only 63% of cells treated with sample CdS/HNTs were as dead/membrane compromised [23].” why did you use reference for this sentence while it is your results?
- Why did you choose a type of cancer cell line for cytotoxicity assay?
- “This phenomenon indicates the decrease of cell enzymatic activity after treatment with this sample without the losing of membrane integrity. Based on our data it can be concluded that compared with two conventional cell viability assays, metabolic properties were more sensitive and accurate for characterizing cellular responses to samples. MTT- test revealed that toxic compare to other samples.” what do you mean?
- Please check the effect of samples on the amount of ROS production.
Author Response
- Please check the whole text, there are some grammatical mistakes in the text that should be corrected.
We checked the whole text and corrected mistakes.
- Please rewrite material section.
Done
- Please explain about the results of figure 1B, STEM results.
According to the results of electron microscopy investigations (Figure 2 and Figure 3 A) all samples have different shape and size of the carriers. The HNTs (in CdS/HNTs) length varies from 200 to 700 nm and diameter is from 7 to 15 nm. CdS/MCM-41 was characterized by MCM-41 microspheres and amorphous MCM-41 with particle size from 600 nm to 1.2 µm. In CdS/MCM-41/HNTs the morphology of carrier represented a composite where MCM-41 was formed mostly on the surface of clay nanotubes and the size of such particles was 200-700 nm in length and 40-200 nm in width. SBA-15 and SBA-15-SH were characterized by mostly cylindric particles with length of up to 1.2 µm (Figure 2 A e). The description was added to page 5 lines 201-211.
- “Cd content revealed by ICP-EOS was 4.6 wt% for CdS/MCM-41, 4.5 wt% for CdS/SBA-15, 4.7 wt% for CdS/SBA-15-SH, 4.3 wt % for CdS/MCM-41-HNTs, 4.1 wt% for CdS/HNTs. The Zeta-potential values of CdS/HNTs, CdS/MCM-41, CdS/MCM-41-HNTs, CdS/SBA-15, CdS/SBA-15-SH are -20, -34, -50, -45, -48 mV respectively” please explain more about these results. Moreover, please write the results of Zeta potential tests in a table.
We added table 1.
The discussion of the results was added to the discussion section lines 306-314.
- “. Bang gap energy calculated using Tauc method equal to 2.36 eV for CdS/MCM-41 which is close to bulk CdS (м.б. написать сколько это?), 2.45 eV for CdS/SBA-15, 2.47 eV for CdS/SBA-15-SH, 2.54 eV for CdS/MCM-41-HNTs and 2.58 eV for CdS/HNTs.” please explain about these results more. What did they show?
The discussion of the results was added to the discussion section lines 316-317, 320-323.
- Authors didn’t mention the results of XRD tests.
We added figure 1 as well as the description of results to the text, page 4, lines 179-185, 307-308. Crystalline structure of halloysite nanotubes in the sample CdS/HNTs was revealed. The characteristic signals on low angles proved the preservation of MCM-41 and SBA-15 pore structure after CdS synthesis in CdS/MCM-41, CdS/MCM-41-HNTs, CdS/SBA-15, CdS/SBA-15-SH (Figure 1). Signals correspondent to CdS at 26.7°, 44.1° ,52.2° were detected on refractograms of all samples except for CdS/SBA-15-SH. In case of CdS/HNTs reliable signal at 44.1° was observed.
- “For all other samples 100% bacterial growth inhibition was observed for the concentrations in 125-1000 µg/mL range. In some cases, even a small concentration of composite demonstrated strong bactericidal effect. Thus, for instance, 63 µg/mL CdS/MCM-41/HNTs and CdS/SBA-15 (ca. 3.8 µg/mL of CdS) showed 50% S. aureus growth inhibition efficiency under visible light.” why it is like that? What does this show?
The discussion was added on page 12 lines 335-340, 341-351.
- Please redraw the curve of MTT assay based on the cell viability percentage and explain the results of that.
Done
- “Only 63% of cells treated with sample CdS/HNTs were as dead/membrane compromised [23].” why did you use reference for this sentence while it is your results?
We corrected this.
- Why did you choose a type of cancer cell line for cytotoxicity assay?
Human lung carcinoma cell culture A549 is widely used as a model for studying the toxicity of various classes of compounds, including nanomaterials. In addition, this cell type has a number of metabolic pathways that ensure the insensitivity of these cells to toxicants, if materials exhibit a toxic effect on this model, then this effect will be even more pronounced on normal non-malignant cells. The discussion was added on page 8, lines 275-279.
- “This phenomenon indicates the decrease of cell enzymatic activity after treatment with this sample without the losing of membrane integrity. Based on our data it can be concluded that compared with two conventional cell viability assays, metabolic properties were more sensitive and accurate for characterizing cellular responses to samples. MTT- test revealed that toxic compare to other samples.” what do you mean?
We rewrote this section (page 11, lines 358-369).
- Please check the effect of samples on the amount of ROS production.
Unfortunately, we could not conduct the experiment in the short period given for a revision. We believe that this is an object for a separate study.
Reviewer 4 Report
In the manuscript, CdS nanoparticles were synthesized on the surface of natural and synthetic mesoporous aluminosilicates and silicates and used to study the photocatalytic degradation of S. aureus under visible light. However, from my perspective, the manuscript needs to be carefully revised and supplemented before acceptance.
1. Three are some grammar and format errors in this manuscript. Language needs substantial improvement. Please consult a native English speaker or a language editing service. Here are some examples:
(1) “that” in Line 45, Page1 should be “than”
(2) the sentence “To obtain MCM-41/HNTs the halloysite nanotubes were added to CTAB solution to obtain MCM-41 to HNTs ratio equal to 1” in Line 82, Page 2 doesn't read smoothly.
(3) “S. aureus” in Line134 and 135, Page 3 should be “S. aureus”. And the“Staphylococcus aureus” should be used for the first time.
(4) “are” in Line 196 should be “were”
(5) there are some errors in Line 124 and Line 203
2. It is difficult to identify whether the nanomaterials have been synthesized successfully just by the characterization in the manuscript. So more characterizations such as FT-IR spectra should be added to characterize the successful synthesis of samples. (for example, SBA-15-SH/SBA-15)
3. Surface area and pore structure analysis by BET should be added.
4. In Figure 1C, it is better to use one color to represent the same element in the EDX mapping.
5. The S. aureus was used to verify the antibacterial activity of CdS nanoparticles stabilized on different mesoporous carriers. How about Gram-negative bacterium such as E.coli?
Author Response
Three are some grammar and format errors in this manuscript. Language needs substantial improvement. Please consult a native English speaker or a language editing service. Here are some examples:
- (1) “that” in Line 45, Page1 should be “than”
We corrected it
- (2) the sentence “To obtain MCM-41/HNTs the halloysite nanotubes were added to CTAB solution to obtain MCM-41 to HNTs ratio equal to 1” in Line 82, Page 2 doesn't read smoothly.
We corrected it
- (3) “S. aureus” in Line134 and 135, Page 3 should be “ aureus”. And the“Staphylococcus aureus” should be used for the first time.
Corrected. Lines 62-63.
- (4) “are” in Line 196 should be “were”
corrected
- (5) there are some errors in Line 124 and Line 203
corrected
- It is difficult to identify whether the nanomaterials have been synthesized successfully just by the characterization in the manuscript. So more characterizations such as FT-IR spectra should be added to characterize the successful synthesis of samples. (for example, SBA-15-SH/SBA-15)
We conducted the FT-IR spectroscopic analysis (figure 4) as well as XRD (Figure 1). Discussion was added to the text (page 4, lines 179-185, page 8 lines 230-236,page 11, lines 306-315).
- Surface area and pore structure analysis by BET should be added.
We performed surface area and pore structure analysis by BET for samples CdS/HNTs, CdS/MCM/HNTs and CdS/SBA-15, surface area values were added to the Table 1.
- In Figure 1C, it is better to use one color to represent the same element in the EDX mapping.
We corrected this.
- The S. aureus was used to verify the antibacterial activity of CdS nanoparticles stabilized on different mesoporous carriers. How about Gram-negative bacterium such as E.coli?
We added the data on E.coli to the text (page 10, lines 270-272, page 12 338-340, 354-358)
Round 2
Reviewer 2 Report
- The format of the references is not uniform. In some reference titles, only the first letter of the first word is capitalized, and in some references, the first letter of all words is capitalized. So you need to carefully check the format of the revised reference. For example, referene 7, 10, 11, 12, 14, 15, etc.
Author Response
- The format of the references is not uniform. In some reference titles, only the first letter of the first word is capitalized, and in some references, the first letter of all words is capitalized. So you need to carefully check the format of the revised reference. For example, referene 7, 10, 11, 12, 14, 15, etc.
Answer:
Thank you for the comments. We corrected this.
Reviewer 3 Report
Thanks for addressing the comments.
Author Response
Thanks you very much for the opportunity to improve our work.
Reviewer 4 Report
Figure 4 was not uploaded into the manuscript.
Author Response
Figure 4 was not uploaded into the manuscript.
Answer
We uploaded it.